# When Protein Dynamics Matter: Integrating Molecular Dynamics into Protein Foundation Models

**Reem Aboelsoud, Javad Kasravi, Stefan Kesselheim & Alina Bazarova**
Jülich Supercomputing Centre
Forschungszentrum Jülich
Jülich, 52428, Germany
{r.aboelsoud,j.kasravi,s.kesselheim,al.bazarova}@fz-juelich.de
Helmholtz AI, Germany

## Abstract

Proteins are dynamic molecules whose function depends not only on sequence and structure but also on conformational changes over time. We investigate how Molecular Dynamics (MD) trajectories can be integrated as an additional modality in protein foundation models by extending OneProt with all-atom, time-resolved MD data curated from mdCATH, GPCRmd, and ATLAS databases. These trajectories encode protein flexibility, conformational variability, and thermodynamic sensitivity, complementing static sequence-, structure-, and text-based representations. Using a pre-trained transformer-based MDGen encoder, we perform systematic pre-training ablations and evaluate the resulting representations across diverse protein prediction tasks. We find that incorporating MD trajectories consistently improves performance on downstream tasks sensitive to dynamics or structural context, particularly when explicit structural information is limited. Our results demonstrate that transformer-based MD encoders capture biologically meaningful dynamic signals that enhance protein foundation models, highlighting the value of integrating protein dynamics for potential applications such as protein engineering and drug discovery.

## 1 Introduction

Recent advances in machine learning have enabled multi-modal foundation models that integrate diverse biological information to improve predictive performance across a range of protein understanding tasks.

While sequence and structure are now well-studied and widely available at scale, Molecular Dynamics (MD) trajectories represent an under-exploited, fundamentally different modality that captures temporal conformational behavior beyond static representations. MD simulations can, in principle, reveal flexibility, transitions between functional states, and fine-grained interaction dynamics that are not visible from sequence or a single static structure alone. However, leveraging MD data in machine learning presents substantial challenges: the data are produced under heterogeneous simulation conditions, varying force fields, temperatures, and lengths; large volumes of simulation data often remain siloed in individual laboratories rather than published in standardized databases. Unlike sequence and structure data, which have large curated repositories, MD simulations lack a universally adopted, comprehensive public database, and even when available, the sheer diversity of formats and sampling protocols complicates unified modeling efforts, Tiemann et al. (2024); Abraham et al. (2019); Roy et al. (2024). Moreover, it remains unclear to what extent generic MD data contain task-relevant information versus noise, for broad protein modeling objectives. These limitations motivate examining whether MD trajectories provide complementary information beyond what is captured by sequence and structure, thereby suggesting their integration into multi-modal representation learning frameworks.

Architectures such as ImageBind pioneered joint embedding spaces that align heterogeneous modalities of the same object, enabling information exchange across modalities within a shared representation space in other domains Girdhar et al. (2023), motivating similar approaches for biological data.

Multimodal protein language models that integrate both sequence and structural information consistently outperform sequence-only baselines, with different architectures offering complementary advantages. Early fusion methods that jointly encode sequence and structure improve functional prediction on benchmarks such as Gene Ontology, Kulmanov & Hoehndorf (2020); Gligorijević et al. (2021). Structure-aware models like MULAN, which enrich sequence embeddings with angle-based structural features, better capture 3D constraints relevant to protein function, Frolova et al. (2025). Similarly, token-level multi-modal frameworks such as SST-ResNet show that jointly encoding sequence–structure tokens enhances functional annotation performance, Zhou et al. (2025). Beyond sequence and structure, recent models incorporate textual information: ProGraphTrans combines graph-based structural context with semantic functional annotations to yield more informative protein representations, Zeng et al. (2025). Extending this idea further, tri-modal systems such as ProTrek, inspired by ImageBind, align sequence, structure, and functional text in a shared embedding space to provide unified multi-modal protein representations Su et al. (2024). OneProt, also

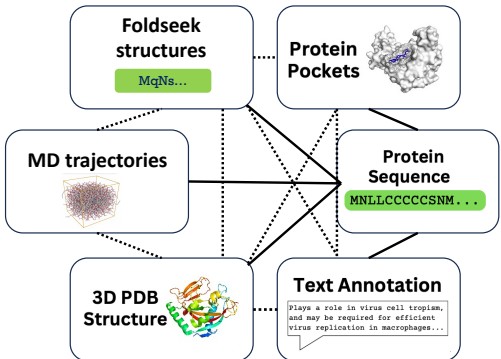

Figure 1: **Overview of the OneProt modality alignment scheme.** Solid lines indicate modality pairs explicitly aligned to the sequence during training, while dotted lines show indirect cross-modal alignment that emerges via the sequence without direct training.

based on ImageBind, is a modular multi-modal foundation model for proteins that aligns diverse data types, including sequence, structure, represented as both graphs and foldseek tokens van Kempen et al. (2024), binding sites, and textual annotations, into a shared latent space via self-supervised contrastive alignment, Flöge et al. (2025). Instead of requiring full multi-way modality matching, OneProt performs pairwise alignment using sequence as an anchor, allowing efficient training with incomplete modality coverage, Figure 1. The projection head maps each modality into a shared embedding space and is updated during training, while pre-trained encoders can be frozen or LoRa fine-tuned Hu et al. (2021), keeping the model lightweight in trainable parameters. For downstream tasks, enriched modality embeddings from the joint representation space are extracted from the pre-trained backbone and passed to a lightweight Multi-layer Perceptron (MLP) projection head for supervised fine-tuning. This design leads to competitive performance across diverse downstream tasks and, due to its modularity, facilitates ablation studies as well as the integration of additional modalities.

Therefore, by evaluating the contribution of the MD modality, we can rigorously assess when and how temporal dynamics add value, clarifying their role in multi-modal protein foundations and establishing principles towards incorporating MD information into general protein representation learning.

In this manuscript, we describe how OneProt framework can be extended with an additional MD modality, detail the data curation process, and present ablation studies alongside statistical comparisons of the resulting experiments on a set of downstream tasks.

## 2 MATERIALS AND METHODS

### 2.1 REPRESENTATION ALIGNMENT

OneProt is trained on paired multi-modal protein data, with each modality providing complementary information. All modalities are aligned per protein, such that data from different modalities correspond to the same biological entity. For each modality pair $(\mathbf{F}, \mathbf{E})$, $\mathbf{F}$ represents the protein sequence, and $\mathbf{E}$ denotes the paired modality. A sample pair $(a_i, b_i)$ consists of two data points from the same protein across different modalities, with $a_i$ representing the sequence. Each modality has its own encoder $\phi$ that generates the corresponding embeddings: $\boldsymbol{a_i'} = \phi_\mathbf{F}(a_i)$ for the sequence and $\boldsymbol{b_i'} = \phi_\mathbf{E}(b_i)$ for the other modality, where $\phi_\mathbf{F} : \mathbf{F} \to \mathbb{R}^k$ and $\phi_\mathbf{E} : \mathbf{E} \to \mathbb{R}^l$ with $k$ and $l$ potentially differing. The representations $(a_i', b_i')$ are then mapped into the shared embedding space $\mathbb{R}^m$ using MLP projections $proj$ and subsequently $L_2$-normalized, yielding the final embeddings

$$(\boldsymbol{a_i}, \boldsymbol{b_i}) = (proj_\mathbf{F}(\boldsymbol{a_i'})/||proj_\mathbf{F}(\boldsymbol{a_i'})||_2, proj_\mathcal{E}(\boldsymbol{b_i'})/||proj_\mathbf{E}(\boldsymbol{b_i'})||_2)$$

We aim to align embeddings of positive modality pairs $(a_i, b_i)$, while pushing apart embeddings of negative pairs $((a_i, b_j)$ for $i \neq j$, following the cross-modal contrastive learning framework of CLIP Radford et al. (2021) and using the approach of aligning latent representations. To achieve this, we apply the InfoNCE contrastive loss van den Oord et al. (2018) over a batch of modalities $\{(a_1, b_1), \ldots, (a_n, b_n)\}$ defined as follows:

$$L_{\mathbf{F},\mathbf{E}} = -\frac{1}{n}\sum_{i=1}^{n}\log\frac{\exp(\boldsymbol{a_i}^\top \boldsymbol{b_i}/\tau)}{\exp(\boldsymbol{a_i}^\top \boldsymbol{b_i}/\tau) + \sum_{j\neq i}\exp(\boldsymbol{a_i}^\top \boldsymbol{b_j}/\tau)}, \tag{1}$$

where $\tau$ is the temperature parameter, set to 1 in our case. The total loss used to train OneProt when aligning two modalities is defined as:

$$L_{\text{total}} = L_{\mathbf{F},\mathbf{E}} + L_{\mathbf{E},\mathbf{F}}, \tag{2}$$

This symmetric loss enforces the model to align embeddings bidirectionally, enabling retrieval from $\mathbf{F}$ to $\mathbf{E}$ and vice versa, consistent with CLIP-style cross-modal contrastive approach. Training is then carried out according to the OneProt framework.

### 2.2 MODEL ARCHITECTURE

We adopt the original modular OneProt architecture, which integrates five modalities using a combination of GNN and Transformer encoders, and extend it with an additional MD modality by incorporating the latent MD encoder from MDGen model Jing et al. (2024). In contrast to conventional MD approaches that simulate physical trajectories using physics-based integrators, MDGen is a deep generative model that learns to represent and generate full MD trajectories as sequences of 3D molecular structures. It tokenizes time-series MD data into $SE(3)$-invariant representations based on roto-translation offsets and torsion angles, and learns the joint distribution over these trajectory tokens so that it can efficiently reproduce or interpolate dynamic behavior such as forward simulation, trajectory upsampling, and conditional inpainting of missing segments. More specifically, MD trajectories are represented as a series of $SE(3)$-invariant tokens with dimensions $T \times L$, where $T$ is the number of frames and $L$ is the number of residues, enabling the model to capture the spatiotemporal dynamics of molecular motion directly from data.

These latent embeddings provide a compact representation of dynamic conformational changes that can be integrated into multi-modal frameworks such as OneProt for downstream prediction or design tasks. We emphasize that the latent MD encoder corresponds to the representation-learning component of the original MDGen architecture, which, in its full form, uses a flow-based generative model to learn and generate MD trajectories. Here we use only the encoder and do not employ its generative sampling mechanism.

The latent MD encoder is based on a Scalable Interpolant Transformer Ma et al. (2024), with time-wise attention replaced by the long-context Hyena architecture Poli et al. (2023), enabling efficient modeling of long temporal contexts. By inheriting this backbone, our encoder extracts latent features

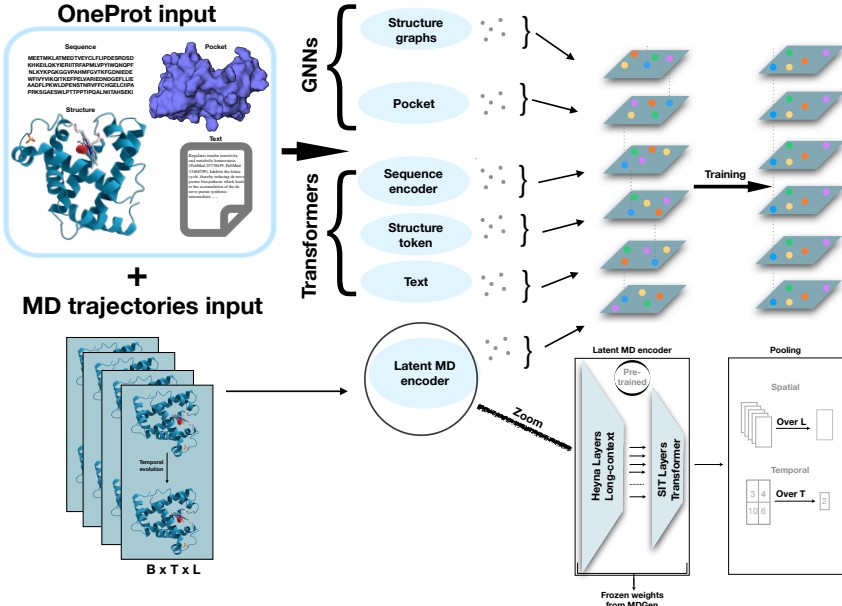

Figure 2: **Extension of OneProt by incorporating MD trajectories.** MD trajectories are incorporated alongside its five existing modalities: sequence and text (transformers), 3D structure (transformer + GNN), and binding pockets (GNN). The inset shows the MD encoder, capturing temporal dynamics with long-context Hyena layers and spatial features with SIT layers, pooled into a fixed-length representation for the shared latent space.

that capture both protein conformations and their temporal evolution from MD trajectories, Figure 2. We use the default MDGen hyperparameter settings and load the weights from the paper provided in the corresponding repository.

Given an input trajectory batch, the Latent MDGen encoder produces hidden representations of shape $H \in \mathbb{R}^{B \times T \times L \times D}$, where $B$ is the batch size, $T$ the number of frames, $L$ the number of residues, and $D$ the latent hidden dimension.

To obtain a fixed-length trajectory embedding compatible with OneProt's multimodal framework, we apply temporal pooling over frames followed by spatial pooling over residues. The pooled MD representations are then projected into the shared OneProt space using an MLP, producing 1024-dimensional embeddings, consistent with the remaining OneProt modalities.

## 2.3 PRE-TRAINING ABLATIONS

Following the downstream evaluation protocols established in OneProt, we conducted ablations that previously yielded the most significant insights, extending them with the additional MD modality. We excluded models lacking a pretrained text encoder, as these configurations exhibited poor performance in prior experiments and showed limited contribution to downstream tasks. We also observed that the presence of at least one structure modality, whether encoded by a Graph Neural Network (GNN) or Transformer model, was critical for strong performance, as demonstrated in the OneProt ablation study. Consequently, we focused on model variants comprising text and at least one structural modality, and we omitted the GNN structure+text ablation due to its weak preliminary performance. Models were pretrained for 32,900 optimizer steps using distributed data parallel training across 128 A100 GPUs of JUWELS Booster Supercomputer, Kesselheim et al. (2021). During pre-training, the latent MD encoder was kept frozen, while only the weights of the MLP projection were updated. MD batches were sampled independently with a small batch size of 2, reflecting the increased computational cost of processing trajectories. Batch sizes and other hyperparameters for the remaining modalities were kept consistent with the original OneProt configuration.

## 2.4 DOWNSTREAM TASKS AND PERFORMANCE EVALUATION

We follow the downstream evaluation protocol established in OneProt to assess the transferability of pretrained multimodal protein representations. Our evaluation covers a diverse suite of tasks spanning protein stability, interactions, subcellular localization, metal binding, and functional annotation, using the same benchmark datasets and metrics as in the original study. Thermostability is assessed as a regression task using Spearman's rank correlation coefficient $\rho$ between predicted and experimentally measured stability values from the Meltome Atlas, capturing how well the model predicts resistance to thermal unfolding Dallago et al. (2021); Jarzab et al. (2020). Human Protein–Protein Interaction (HumanPPI) is a binary classification task where positive pairs are experimentally validated interactions from curated databases, and performance is measured by accuracy Xu et al. (2022). Metal Ion Binding predicts whether a protein binds metal ions based on experimentally annotated binding sites in the Protein Data Bank (PDB), evaluated by classification accuracy Hu et al. (2022); Burley et al. (2023). Subcellular localization uses the DeepLoc benchmarks, with DeepLoc2 as a binary (membrane vs. soluble) and DeepLoc10 as a ten-class task, both evaluated by accuracy on experimentally annotated localizations from UniProt Almagro Armenteros et al. (2017); Boutet et al. (2007). Enzyme Commission (EC) number prediction and Gene Ontology (GO) annotation (MF: molecular function, BP: biological process, CC: and cellular component) are multilabel binary classification tasks evaluated using the maximum $F_1$ score ($F_{max}$), requiring recognition of conserved motifs and functional properties Gligorijević et al. (2021). Collectively, these benchmarks measure how effectively the pretrained multimodal representations transfer to biologically relevant sequence, structure, interaction, and functional prediction problems.

By using the same downstream tasks, datasets, and evaluation metrics introduced in OneProt, we ensure that observed performance differences can be attributed to the inclusion of the MD modality. For each task, we train a task-specific MLP on the embeddings extracted from the pretrained OneProt+MD model, in order to predict the corresponding labels. To account for variability, each model-task combination was evaluated over six independent runs with different random initializations, using the same hyperparameter sweep as in OneProt. For each run, we recorded the maximum validation metric and reported the mean of the corresponding six values on the test set. To assess the impact of the MD modality, we compared OneProt + MD against the original OneProt (without MD) using a Wilcoxon rank sum test (two-sided), to evaluate whether adding MD resulted in statistically significant improvements in the corresponding evaluation metrics.

## 2.5 DATASET PREPARATION

We use five primary data modalities: protein sequences, structural representations (graphs and tokens), binding site annotations, textual annotations inherited from OneProt Flöge et al. (2025), and molecular dynamics trajectories. For clarity, we adopt the following abbreviations for different modality combinations throughout this paper: OneProt-5 denotes the full five-modality model (sequence, pocket, text, and structural representations as graphs and tokens); OneProt-4 includes sequence, pocket, text, and graph-based structure; Text+ST+Pocket comprises sequence, pocket, text, and structure tokens; Text+ST corresponds to sequence, text, and structure tokens; and Text refers to the model using only sequence and text modalities.

**Protein sequences.** Present in all training pairs, represented as amino acid strings and tokenized using the ESM2-650M model Lin et al. (2022), comprising 1.04M datapoints.

**Structural data.** Encoded using two complementary representations:

  i. **Structure graphs.** Proteins are represented as 3D graphs with amino acid nodes and edges encoding spatial and chemical interactions, processed using the GNN model ProNet Wang et al. (2023), comprising 656K datapoints.
  ii. **Structure tokens.** Discrete foldseek tokens van Kempen et al. (2024), capturing geometric and conformational information for Transformer-based encoding using ESM-35M Lin et al. (2022), comprising 1M datapoints.

**Binding site data.** Residue-level annotations describing protein–ligand interaction regions, represented as graphs and encoded by ProNet Wang et al. (2023) comprising 341K datapoints.

**Textual data.**    Provides biological and functional context using curated UniProt annotations, encoded with MSR BiomedBERT Gu et al. (2021), encompassing 546K datapoints.

**Molecular dynamics trajectories.**    To incorporate protein dynamics into OneProt, we curated trajectories from three MD datasets: mdCATH Mirarchi et al. (2024), GPCRmd Rodríguez-Espigares et al. (2020), and ATLAS Vander Meersche et al. (2023), all available through MDRepo Roy et al. (2024). These datasets provide complementary coverage of protein families, structural classes, and timescales.

mdCATH contains MD simulations of representative protein domains from the CATH structural classification Waman et al. (2025), simulated with a classical force field across five temperatures and five replicates, recording coordinates and forces every 1 ns for trajectories up to 500 ns. GPCRmd provides standardized MD trajectories of G-protein-coupled receptors, with many systems simulated in triplicate for up to 500 ns per replicate and frames saved every 0.2 ns, capturing conformational landscapes relevant to receptor activation and signaling. ATLAS consists of standardized all-atom molecular dynamics trajectories for a large set of proteins selected to represent the structural diversity of the PDB. For each protein it provides three replicates of 100 ns simulations with frames saved every 10 ps, enabling comparative analysis of protein flexibility across folds and sizes.

To standardize temporal resolution across these heterogeneous sources, we upsampled GPCRmd and ATLAS trajectories to mdCATH's 1 ns spacing (every 5 frames for GPCRmd and every 100 frames for ATLAS), filtered for trajectories longer than 100 frames, and sampled fixed-length trajectory segments with random start positions. For mdCATH, we used only the trajectories sampled at 320 K. This amounted to overall 5005 unique protein sequences.

For trajectories longer than 100 frames, a random starting point was sampled for each segment so the model could learn diverse portions of the simulation. When multiple simulation replicas were available for a protein, one replica was randomly selected per sample. Cartesian coordinates were then converted into torsion angles and rigid-frame representations compatible with the latent MD encoder's input format.

Pretraining data splits followed the OneProt protocol, ensuring that sequences in different splits shared no more than 50% sequence identity. We also ensured that there is no data leakage between the newly added MD data and the downstream evaluation datasets.

## 3    RESULTS

Tables 1 and A.1 compare the performance of our five model variants incorporating MD data with the original OneProt results, which do not include MD data.

The Thermostability task shows a consistent statistically significant improvement through all models variants when combined with MD ($p \leq 0.01$), with Text + MD encoder corresponding to the lowest performance.

For the binary subcellular localization task (DeepLoc2), models with partially removed structural information, OneProt-4, Text+ST+Pocket, and Text+ST, show statistically significant improvement ($p < 0.05$), with the Text+ST model achieving the highest accuracy. In contrast, for the multiclass formulation (DeepLoc10, Table A.1), adding MD benefits only the model lacking high-level structural abstractions encoded by structure tokens (OneProt-4), while the Text-only model achieves the highest performance regardless of MD inclusion.

For the HumanPPI task, the lowest $p$-values are observed for models combining text with structural tokens, OneProt-5, Text+ST, and Text+ST+Pocket ($p < 0.03$). The Text and OneProt-4 models combined with MD also show positive trends in performance, but these improvements are not statistically significant.

For the Metal Ion Binding task, models without graph-based coordinates (Text+ST, Text+ST+Pocket, Text) exhibit significant improvement with inclusion of MD ($p < 0.006$), whereas models incorporating GNN-encoded structures remain largely unaffected.

Performance on the EC task is largely insensitive to MD inclusion (Table A.1). For GO tasks, only GO-CC shows significant improvement with MD across nearly all model variants ($p \leq 0.04$),

even without explicit structural representation. However, when all five modalities are combined in OneProt-5, MD provides no statistically significant benefit ($p = 0.552$). For GO-BP task, text-only models already perform strongly, with the Text model (without MD data) achieving the best overall performance. Nevertheless, incorporating MD data provides small but in some cases statistically significant improvements. OneProt-5 and Text+ST models show significant gain after integrating MD data ($p < 0.01$). Performance on the GO-MF task is largely unaffected by the inclusion of MD data. Only Text+ST+Pocket variant demonstrates statistically significant improvement with MD incorporation ($p = 0.037$).

| Task/Metric | Model | With MD (Mean $\pm$ Std) | Without MD (Mean $\pm$ Std) | P-Value |
|---|---|---|---|---|
| DeepLoc2
Accuracy | OneProt-5 | $0.925 \pm 0.002$ | $0.924 \pm 0.002$ | 0.128 |
| | OneProt-4 | $0.926 \pm 0.000$ | $0.921 \pm 0.003$ | **0.045** |
| | Text+ST | $0.934 \pm 0.001$ | $0.928 \pm 0.003$ | **0.004** |
| | Text+ST+Pocket | $0.927 \pm 0.002$ | $0.913 \pm 0.003$ | **0.004** |
| | Text | $0.929 \pm 0.001$ | $0.929 \pm 0.002$ | 1.000 |
| MetalIonBinding
Accuracy | OneProt-5 | $0.762 \pm 0.003$ | $0.762 \pm 0.016$ | 0.810 |
| | OneProt-4 | $0.772 \pm 0.003$ | $0.773 \pm 0.005$ | 0.575 |
| | Text+ST | $0.768 \pm 0.005$ | $0.747 \pm 0.011$ | **0.004** |
| | Text+ST+Pocket | $0.767 \pm 0.008$ | $0.745 \pm 0.010$ | **0.006** |
| | Text | $0.772 \pm 0.003$ | $0.740 \pm 0.022$ | **0.004** |
| ThermoStability
Spearman's $\rho$ | OneProt-5 | $0.687 \pm 0.002$ | $0.673 \pm 0.010$ | **0.004** |
| | OneProt-4 | $0.684 \pm 0.001$ | $0.668 \pm 0.006$ | **0.004** |
| | Text+ST | $0.687 \pm 0.000$ | $0.666 \pm 0.014$ | **0.004** |
| | Text+ST+Pocket | $0.686 \pm 0.004$ | $0.670 \pm 0.006$ | **0.010** |
| | Text | $0.678 \pm 0.003$ | $0.656 \pm 0.005$ | **0.005** |
| HumanPPI
Accuracy | OneProt-5 | $0.882 \pm 0.006$ | $0.859 \pm 0.021$ | **0.024** |
| | OneProt-4 | $0.896 \pm 0.006$ | $0.888 \pm 0.017$ | 0.522 |
| | Text+ST | $0.883 \pm 0.009$ | $0.845 \pm 0.012$ | **0.004** |
| | Text+ST+Pocket | $0.896 \pm 0.010$ | $0.859 \pm 0.016$ | **0.008** |
| | Text | $0.881 \pm 0.008$ | $0.853 \pm 0.013$ | 0.471 |
| GO-BP
$F_{max}$ | OneProt-5 | $0.496 \pm 0.001$ | $0.492 \pm 0.003$ | **0.010** |
| | OneProt-4 | $0.499 \pm 0.001$ | $0.495 \pm 0.003$ | 0.066 |
| | Text+ST | $0.499 \pm 0.001$ | $0.496 \pm 0.003$ | **0.004** |
| | Text+ST+Pocket | $0.499 \pm 0.001$ | $0.497 \pm 0.004$ | 0.093 |
| | Text | $0.500 \pm 0.001$ | $0.503 \pm 0.002$ | 0.066 |
| GO-CC
$F_{max}$ | OneProt-5 | $0.560 \pm 0.001$ | $0.556 \pm 0.005$ | 0.522 |
| | OneProt-4 | $0.564 \pm 0.002$ | $0.555 \pm 0.006$ | **0.037** |
| | Text+ST | $0.564 \pm 0.001$ | $0.549 \pm 0.007$ | **0.004** |
| | Text+ST+Pocket | $0.565 \pm 0.003$ | $0.546 \pm 0.004$ | **0.004** |
| | Text | $0.568 \pm 0.002$ | $0.561 \pm 0.005$ | **0.008** |
| GO-MF
$F_{max}$ | OneProt-5 | $0.658 \pm 0.002$ | $0.656 \pm 0.002$ | 0.093 |
| | OneProt-4 | $0.657 \pm 0.002$ | $0.656 \pm 0.001$ | 0.128 |
| | Text+ST | $0.657 \pm 0.001$ | $0.661 \pm 0.002$ | 0.093 |
| | Text+ST+Pocket | $0.662 \pm 0.003$ | $0.657 \pm 0.005$ | **0.037** |
| | Text | $0.657 \pm 0.001$ | $0.656 \pm 0.004$ | 1.000 |

Table 1: Task-wise model comparison showing Mean $\pm$ Standard Deviation over six independent downstream runs. Bold $p$-values of indicate $p < 0.05$. The first column shows the task and the corresponding evaluation metric.

## 4 DISCUSSION

In this manuscript, we assessed the impact of MD trajectories on downstream performance within the OneProt framework. Our results show that they provide a complementary modality for multi-modal

protein foundation models, leading to notable improvements across tasks, particularly when static structural or textual representations alone are insufficient to capture aspects of protein behavior. These findings indicate that MD trajectories encode rich, biologically meaningful information that enhances protein representations.

In particular, the improvement on the thermostability task likely arises because MD captures conformational flexibility and dynamic fluctuations, providing insights into states relevant for structural stability, as highlighted in Dror & et al. (2012); Gómez-Flores et al. (2023); Kuzmanić et al. (2020); Fowler & Williamson (2022).

For a binary subcellular localization task of discriminating between membrane and soluble proteins (DeepLoc2), MD becomes critical when explicit structural information is not fully present. In the Text+ST+Pocket model, lacking graph-based 3D coordinates, MD allows the model to learn implicit structural and dynamic features from conformational ensembles, Shaw & et al. (2010); Mirarchi et al. (2024). In OneProt-4, MD complements graph-based encodings by providing time-resolved information on conformational stability and flexibility. Text+ST model benefits from a strong alignment between UniProt-derived textual annotations and compact structural abstractions that capture global organization. Incorporating MD trajectories enriches the representation with dynamic information on conformational stability and flexibility, key for localization signals such as transmembrane regions, while pocket-level features, focused on ligand binding, are less relevant, allowing Text+ST model to achieve the best performance. In contrast, Text- and OneProt-5 models show little effect from MD: the former lacks structural context to exploit trajectories, while the latter's static modalities already capture relevant information.

In DeepLoc10, a multi-class subcellular localization task, the ten localization categories rely on strong sequence-encoded localization signals, particularly N-terminal targeting motifs, which are well captured by sequence features and UniProt-derived annotations. As a result, the Text-only model achieves the highest performance, and additional structural or dynamic information provides little benefit.

For Human Protein-Protein Interactions task, the largest benefit from MD is observed when dynamic information complements both textual descriptions and structure-token representations, reflecting a strong synergy between these modalities. Text provides high-level functional context, structure tokens encode global spatial organization, and MD trajectories expose interaction-relevant conformational ensembles, allowing dynamic information to be interpreted within a biologically meaningful structural framework. In contrast, Text-only models show only non-significant gains from MD, indicating that without explicit structural context the model cannot fully exploit dynamic information. Models combining GNN-based structures with text likewise exhibit modest improvement trends; however, their strong baseline performance limits the measurable impact of additional dynamic information.

For the MetalIonBinding task, MD data is most beneficial for models that lack explicit graph-based coordinates, where precise spatial relationships are otherwise difficult to infer. Metal ion binding is highly sensitive to the geometry of the binding site and the coordinated arrangement of surrounding residues. In this setting, MD trajectories provide complementary information by revealing how coordinating residues move, interact, and maintain spatial proximity over time, thereby compensating for missing structural detail and providing additional performance gains. By contrast, models with GNN-encoded structures already capture the relevant geometric constraints, limiting the additional benefit of MD.

For the EC task, MD has minimal impact, as enzymatic function is largely determined by conserved motifs and active sites van der Weg et al. (2024); Capela et al. (2025); Martin et al. (1998); Lin et al. (2022). These features are primarily represented in sequences and annotations. Since catalytic mechanisms depend on specific residue chemistry and relatively stable structural configurations, the additional dynamic information provides limited signal for EC classification. Similarly, in the TopEnzyme downstream task from the original OneProt study van der Weg & Gohlke (2023), which targets enzyme function using signals related to EC classification, incorporating MD trajectories yielded no statistically significant improvement, consistent with the EC results (data not shown).

GO-CC describes where a protein resides or functions - membrane, cytosol, nucleus, ribosome, or extracellular space —reflecting fundamentally spatial, environment-dependent properties. By capturing time-resolved structural dynamics, MD data reveals protein stability across environments

and the dynamic exposure of localization-relevant regions, explaining the notable performance gains from MD integration. In contrast, GO-MF benefits least from MD data among the three GO tasks. Molecular functions are largely sequence-determined, identifiable from conserved motifs, catalytic residues, or binding-site features. Their localized, chemically specific nature leaves little room for additional insight from global conformational dynamics.

GO-BP task described broader biological roles - signaling, apoptosis, metabolism, transcriptional regulation - which involve conformational switching, conditional activation and allosteric regulation. MD data are well-suited to provide and capture these dynamic features, Ryan V. et al. (2025). However, the biological processes are highly contextual and frequently described in literature annotation, Ryan V. et al. (2025). This explains why the Text model achieves the best model (with or without MD data). On the other hand, we notice that the best performing models combined with MD data are the ones that have, text and structure token representations together. This combination is effective because biological processes are highly contextual, making structural information crucial: it encodes interaction surfaces, domain architecture, and flexibility constraints, which, together with MD-derived conformational states, enhance BP prediction.

Overall, although MD data is available for only 5,005 proteins, each trajectory contains hundreds of frames capturing rich conformational ensembles. Tasks that depend on physical behavior or structure, such as thermostability, protein–protein interactions, metal ion binding, and GO Cellular Component prediction, show clear sensitivity to MD, particularly in models lacking explicit structural representations. In these cases, MD compensates for missing 3D information by encoding residue mobility, spatial density, and transient interactions. However, when no structural context is available at all, as in text-only models, performance improvements from MD may be not as large. Moreover, function-centric tasks gain little from MD overall, as sequence and static structural features already capture the relevant information.

These results suggest that, for applications such as drug discovery, protein engineering, and binding affinity prediction, integrating MD trajectories can significantly enhance models' ability to capture conformational flexibility and dynamic stability, features that are often critical for these tasks but poorly represented by static structures alone. As MD datasets expand, they are likely to play an important role in the next generation of protein foundation models, enabling more accurate, dynamic-aware predictions for tasks where structural dynamics, interactions, and environmental context are key.

### ACKNOWLEDGMENTS

This work is supported by the Helmholtz Association Initiative and Networking Fund in the frame of Helmholtz AI as well as by the Helmholtz Foundation Model Initiative within the project "PROFOUND". The authors gratefully acknowledge the Gauss Centre for Supercomputing e.V. (www.gauss-centre.eu) for funding this project by providing compute time through the John von Neumann Institute for Computing (NIC) on the GCS Supercomputer JUWELS at Jülich Supercomputing Centre (JSC), project id profound.

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

## APPENDIX

| Task | Model | With MD (Mean $\pm$ Std) | Without MD (Mean $\pm$ Std) | P-Value |
|------|-------|--------------------------|------------------------------|---------|
| DeepLoc10 Accuracy | OneProt-5 | $0.805 \pm 0.00016$ | $0.803 \pm 0.002$ | 0.109 |
|  | OneProt-4 | $0.823 \pm 0.00016$ | $0.817 \pm 0.004$ | **0.016** |
|  | Text+ST | $0.812 \pm 0.00019$ | $0.816 \pm 0.004$ | 0.423 |
|  | Text+ST+Pocket | $0.803 \pm 0.00016$ | $0.805 \pm 0.004$ | 0.230 |
|  | Text | $0.826 \pm 0.00015$ | $0.830 \pm 0.004$ | 0.575 |
| EC $F_{max}$ | OneProt-5 | $0.877 \pm 0.001$ | $0.875 \pm 0.005$ | 0.262 |
|  | OneProt-4 | $0.873 \pm 0.001$ | $0.871 \pm 0.003$ | 0.128 |
|  | Text+ST | $0.878 \pm 0.002$ | $0.877 \pm 0.001$ | 0.055 |
|  | Text+ST+Pocket | $0.879 \pm 0.002$ | $0.876 \pm 0.004$ | 0.174 |
|  | Text | $0.875 \pm 0.002$ | $0.876 \pm 0.003$ | 0.631 |

Table A.1: Task-wise model comparison showing Mean $\pm$ Standard Deviation over six independent downstream runs. Bold $p$-values of indicate $p < 0.05$. The first column shows the task and the corresponding evaluation metric.

