# OpenReview forum: "When Protein Dynamics Matter: Integrating Molecular Dynamics into Protein Foundation Models"
_ICLR.cc/2026/Workshop/FM4Science — ICLR 2026 Workshop FM4Science Poster_

### Official Review · Reviewer_s4P3 · 2026-02-22
**An empirical study showing that MD trajectory representations can complement static sequence/structure modalities in protein foundation models, with task-dependent improvements, though limited by small MD data coverage and the absence of dynamics-specific evaluation tasks.**

**Rating:** 6
**Confidence:** 2

**Review:**

This paper extends OneProt by integrating MD trajectories as a sixth modality via a frozen MDGen encoder, evaluating across 10 downstream tasks with rigorous ablation studies and Wilcoxon statistical testing.

Pros:

(1) Well-motivated. Protein dynamics is a genuinely underexplored modality for foundation models.

(2) Systematic ablation with statistical significance testing across multiple model variants.

(3) Thoughtful per-task analysis explaining when and why MD helps (e.g., thermostability, localization) versus when it doesn't (EC, GO-MF).

(4) Careful data curation from three complementary MD databases.

Cons:

(1) Improvements are modest and inconsistent—MD often provides no significant gain when full structural information is present.

(2) Only 5,005 proteins have MD data versus 1M+ for other modalities, limiting generalizability.

(3) MD encoder is frozen with no joint fine-tuning explored.

(4) No dynamics-specific tasks (e.g., conformational change, allostery) where MD should excel most.

---

### Official Review · Reviewer_KeNY · 2026-02-23
**Review on When Protein Dynamics Matter: Integrating Molecular Dynamics into Protein Foundation Models**

**Rating:** 8
**Confidence:** 3

**Review:**

This paper presents a novel approach to protein representation learning by integrating Molecular Dynamics trajectories as a new modality within the multi-modal OneProt foundation model.  The authors utilize a pre-trained latent MD encoder from the MDGen model to capture temporal and spatial conformational changes. They successfully curate a dataset of MD trajectories from mdCATH, GPCRmd, and ATLAS, standardizing the temporal resolution to integrate this dynamic information. Through extensive ablation studies, the authors evaluate the enriched model on multiple downstream tasks. They conclude that MD data consistently improves performance on tasks sensitive to structural context and dynamics, particularly when static structural information is lacking.

A major strength of this work is its robust architecture and the novelty of the modality. While previous foundation models have effectively utilized static sequence and structural data , MD trajectories represent an under-exploited modality capable of revealing transitions between functional states and interaction dynamics that are invisible to static representations.  The methodology smartly leverages OneProt's pairwise alignment, allowing for efficient training anchored to the sequence without requiring full multi-way modality matching. Furthermore, employing the pre-trained MDGen encoder—which utilizes a long-context Hyena architecture—enables the efficient modeling of the long temporal contexts inherent in MD simulations. The evaluation is comprehensive and firmly grounded in biological intuition. The authors perform rigorous, statistically backed evaluations over six independent runs to validate their findings.

There are minor areas that could benefit from future refinement or discussion. For instance, clarifying whether the strategy of applying temporal pooling over frames followed by spatial pooling over residues to create the 1024-dimensional embedding might discard fine-grained spatiotemporal signals would strengthen the methodology section.

Ultimately, this work sets a highly valuable precedent for incorporating temporal dynamics into molecular machine learning.

---

### Meta-Review · Area_Chair_qcfS · 2026-02-28

**Recommendation:** Accept (Poster)
**Confidence:** 4

**Metareview:**

This paper explores protein dynamics as an additional modality for protein foundation models-specifically, the OneProt model. Reviewers found the work well motivated, and the idea of MD trajectories as a modality to be novel. The empirical evidence was sufficient to back the claim that MD trajectories can improve performance in certain contexts (e.g., tasks known to be sensitive to dynamics).  Additional experiments that show the results of ablating aspects of the design and training (e.g., what happens when the latent MD encoder is unfrozen for joint fine-tuning) would strengthen the significance and impact.

---

### Decision · Program_Chairs · 2026-03-03

Accept (Poster)